# New Insights in the Diagnosis of Rare Adenocarcinoma Variants of the Cervix—Case Report and Review of Literature

**DOI:** 10.3390/healthcare10081410

**Published:** 2022-07-28

**Authors:** Cristina Secosan, Oana Balint, Aurora Ilian, Lavinia Balan, Ligia Balulescu, Andrei Motoc, Delia Zahoi, Dorin Grigoras, Laurentiu Pirtea

**Affiliations:** 1Department of Obstetrics and Gynecology, Victor Babes University of Medicine and Pharmacy, 300041 Timisoara, Romania; cristina.secosan@gmail.com (C.S.); aurora1985@yahoo.com (A.I.); bln_lavinia@yahoo.com (L.B.); ligia_balulescu@yahoo.com (L.B.); grigorasdorin@ymail.com (D.G.); laurentiupirtea@gmail.com (L.P.); 2Department of Anatomy and Embryology, Victor Babes University of Medicine and Pharmacy, 300041 Timisoara, Romania; amotoc@umft.ro (A.M.); dzahoi@umft.ro (D.Z.)

**Keywords:** immunohistochemistry, dual-staining, p16/Ki-67, cervix, adenocarcinoma, clear cell, mesonephric

## Abstract

We report the case of a 29-year-old patient with low-grade squamous intraepithelial lesion (L-SIL), negative human papilloma virus (HPV), positive p16/Ki-67 dual-staining and colposcopy suggestive for severe dysplastic lesion. The patient underwent a loop electrosurgical excision procedure (LEEP), the pathology report revealing mesonephric hyperplasia and adenocarcinoma. The patient also opted for non-standard fertility-sparing treatment. The trachelectomy pathology report described a zone of hyperplasia at the limit of resection towards the uterine isthmus. Two supplementary interpretations of the slides and immunohistochemistry (IHC) were performed. The results supported the diagnosis of mesonephric adenocarcinoma, although with difficulty in differentiating it from mesonephric hyperplasia. Given the discordant pathology results that were inconclusive in establishing a precise diagnosis of the lesion and the state of the limits of resection, the patient was referred to a specialist abroad. Furthermore, the additional interpretation of the slides and IHC were performed, the results suggesting a clear cell carcinoma. The positive p16/Ki-67 dual-staining prior to LEEP, the non-specific IHC and the difficulties in establishing a diagnosis made the case interesting. Given the limitations of cytology and the fact that these variants are independent of HPV infection, dual staining p16/Ki-67 could potentially become useful in the diagnosis of rare adenocarcinoma variants of the cervix, however further documentation is required.

## 1. Introduction

A small percentage of cervical cancer such as serous carcinoma (SC), clear cell carcinoma (CCC), mesonephric adenocarcinomas (MSN) and adenocarcinomas exhibiting gastric differentiation (GAS) have been described as non-human papilloma virus (HPV) related to cervical adenocarcinomas. This is due to the very low or no prevalence of HPV infection in these patients (0–30%). Although existing data is scarce regarding the prevalence of HPV in unusual types of cervical carcinoma, it has been stipulated that these types of cervical cancer are unrelated to or independent of HPV infection. Hence, HPV testing is not conclusive in these patients [1,2]. Cytology is also considered to have limited benefits in the diagnosis of rare forms of cervical cancer. Only a very small percentage of patients present an abnormal Pap smear [3,4,5]. On the other hand, p16 staining on the biopsy specimens has revealed diffuse staining not linked to HPV infection in some cases [6,7,8], and Ki-67 can be positive in certain cases regardless of HPV status [9]. Based on the premises that p16 and/or Ki-67 could be positive in the absence of HPV infection, dual-staining could become a useful screening test in patients with rare non-HPV related variants of cervical cancer.

## 2. Case Report

We report the case of a 29-year-old patient whose history of the disease dates back five years prior to diagnosis when she presented a high grade squamous intraepithelial lesion (H-SIL) upon cytological examination. The patient had no history of in utero exposure to diethylstilbestrol (DES). A loop electrosurgical excision procedure (LEEP) was proposed at the time, but the patient refused the procedure and opted for electrocoagulation of the cervical lesion. Despite the recommendation for annual cervical surveillance, the patient received only one Pap smear in the following five years, the result being negative for intraepithelial or malignant lesions (NILM) cytology.

Five years after the H-SIL result, the patient arrived for consultation in our office for the first time. Given her background, a cervical cytology, HPV genotyping by standard DNA and a CINtec test (p16/Ki-67) were performed. The results were L-SIL cytology, negative HPV and positive p16/Ki-67. A colposcopy was performed afterwards and revealed changes suggestive of a severe dysplastic lesion.

The patient underwent LEEP and the pathology report revealed atypical mesonephric hyperplasia with a malignant transformation zone—mesonephric adeno-carcinoma with moderate cell pleomorphism, moderate mitotic activity, without invasion of the lymphovascular space, and resection limits tangential to the lesion (Figure 1a,b).

The abdomino-pelvic magnetic resonance imaging (MRI) carried out after the LEEP identified a 2.2/1.6/1.6 cm formation with suspicion of malignancy which did not exceed the contour of the cervical wall, and no radiologically detectable pelvic lymphadenopathy.

Non-standard fertility sparing treatment was proposed at the MDT (multidisciplinary team meeting), i.e., radical trachelectomy, taking into account the clinical and paraclinical data, namely: 30-year-old nulligesta, nulliparous patient, with desire for fertility preservation; the dimensions of the lesion on the MRI 2.2/1.6/1.6 cm without exceeding the contour of the cervical wall; the pathological report which showed a rare form, but with a weak tendency to aggressiveness (moderate mitotic activity; moderate cell pleomorphism; without invasion of the lymphovascular space). Informed consent was obtained from the patient after explaining the risks and benefits of non-standard fertility- sparing surgical treatment.

Radical vaginal trachelectomy was performed with laparoscopic pelvic lymphadenectomy. The surgery was performed by a surgeon with extensive experience in this type of procedure. The frozen section was performed for the sentinel lymph node and revealed no lymph node invasion.

The pathology report was the following: vaginal fragment:vaginal wall with non-specific chronic inflammation, at the level of the subepithelial stroma we found micro focaries of mesonephric remains within the histological limits of benignity (Figure 2). The lower limit of vaginal resection shows no malignant tumor tissue; uterine isthmus—endocervix—upper limit of resection with benign mesonephric hyperplasia with areas of atypical mesonephric hyperplasia, showing moderate atypia (Figure 3); cervix with previous conization—chronic ulcero-granulomatous cervicitis, condilomatous squamous epithelium; at the level of the subepithelial stroma, there is a tumoral proliferation consisting of delimited tubular structures of cubic cells, non-ciliated, with moderate, eosinophilic or clear cytoplasm, presenting in the lumen an eosinophilic hyaline secretion producing a histological appearance of atypical mesonephric hyperplasia; zone of stromal invasion and malignant transformation—endocervical mesonephric adenocarcinoma with moderate cell pleomorphism and mitotic activity, intraluminal detritus, added inflammation (Figure 4a–c); right ilioobturator lymphadenctomy specimen—eleven lymphonodules with sinus histiocytosis, lipomatosis, no tumor metastasis; left ilioouturator lymphadenctomy specimen—seven lymphonodules with sinus histiocytosis, lipomatosis, no tumor metastasis. The stage according to FIGO classification was pT1b1 N0 Mx.

Faced with this description of a zone of hyperplasia at the limit of resection towards the uterine isthmus, two supplementary interpretations of the slides in two different independent laboratories were requested and an immunohistochemistry was performed, and the immunohistochemistry results are presented in Table 1.

First laboratory: haematoxylin and eosin stain (H&E) staining: Fragments of vaginal and cervical wall with circumferential glandular proliferation especially developed at the level of the cervix, with a predominant tubulocystic pattern. Tubular glandular structures of variable size, bungs, infiltrative, lined with monostratified cubic or cylindrical epithelium, focally flattened, with pale eosinophilic or clear cytoplasm, round or oval nuclei, non-uniform in some areas and with evident pleomorphism in others other areas, irregular nuclear contour, some atypical nuclei, vesicular, with very rare mitoses (2–3/50 large objective microscopic fields). No appearance of perineural or vascular invasion. The lesion was located focally at the level of the vaginal wall. Conclusion: atypical endocervical glandular proliferation, difficult to label as mesonephric adenocarcinoma, associated with large areas of atypical mesonephric hyperplasia, diffuse form; the same type of lesion is present at the level of the vaginal wall and in the deep limit of resection (parameter), R1 LV0. NB: the differential diagnosis between mesonephric hyperplasia and mesonephric adenocarcinoma is difficult to achieve; to correlate with clinical data and re-evaluate in a reference center for gynecological pathology.

Second laboratory: Microscopic description: tissue fragments with appearance of vaginal and cervical wall showing malignant tumor proliferation consisting of agglomerated glandular structures, infiltrative, margined by cubic and rarely cylindrical cells, with moderate pleomorphism, rare mitoses (two to three atypical mitoses/10 HPF), with irregular tachychromia nuclei and some vesicles. The glandular lumen presents homogeneous, eosinophilic amorphous material. The limit of surgical resection is infiltrated by the tumor. Conclusion: the histopathological appearance may be compatible with mesonephric adenocarcinoma; the immunohistochemical aspect orients towards a mesonephric adenocarcinoma.

The abdomino-pelvic MRI performed after surgery (trachelectomy) revealed the modification of the anatomy of the cervix in the postoperative context; area (10–12 mm) at the junction with the uterine body with appearance similar to the lesion described previously (MRI prior to trachelectomy)—tumor remains? No pelvic lymphadenopathy. Moderate fluid accumulations noted in the pouch of Douglas.

Given the discordant pathology reports that were too inconclusive to establish a precise diagnosis of the lesion (hyperplasia or adenocarcinoma) and the state of the limits of resection, the patient and the blocks were referred to a specialist abroad. Supplementary IHC and interpretation of slides was performed in a third and fourth laboratory, with the following results: Belfast Royal Group of Hospitals: Histology of the sections taken from this cervical conization shows unremarkable surface squamous epithelium. Within the underlying tissue, an adenocarcinoma is present predominantly composed of dilated glandular structures lined by attenuated epithelium. Small tubular structures are also present lined by cells with clear cytoplasm and focally with a hobnail appearance. The immunohistochemistry is presented in Table 1. Canonical activating KRAS mutations, NRAS mutations, gain of 1q, no microsatellite instability. TP53 mutations are variably present. Although mesonephric carcinoma comes into the differential diagnosis, given the morphology and the immunophenotype, this represents a clear cell carcinoma. Institut Bergonie, Bordeaux, France: on the examined material we identified an epithelial tumoral proliferation of carcinomatous type, with deep development unrelated to the malpighien epithelial lining. In the connective tissue of the cervix, we identified an adenocarcinoma consisting mainly of dilated glandular structures, bordered by a clarified epithelial lining. There are also small tubular structures bordering an epithelial lining made up of cells with clear cytoplasm and a focally clapboard appearance. No KRAS mutations were present. Although the differential diagnosis arises between a mesonephric tumor, in particular a mesonephric carcinoma, we consider the most likely diagnosis is clear cell carcinoma of the cervix. Of course, an endometrial origin cannot be excluded.

The final diagnosis was therefore clear cell carcinoma and, given the limited reports of successful fertility sparing treatments in the literature, the final multidisciplinary team (MDT) decision was in favor of laparoscopic hysterectomy with vaginal cuff, left adnexectomy and transposition of the right ovary. The investigations, pathophysiological and etiological information including treatment history of the case are presented in Table 2. So far, the patient has 3 years of disease free survival and is under regular monitoring.

## 3. Discussions and Literature Review

### 3.1. Definitions

Cervical adenocarcinomas represent a heterogeneous group of tumors. According to the 2020 WHO Classification, adenocarcinomas of the gastric type, mesonephric type and clear cell type have been classified as HPV-independent cervical adenocarcinomas [10,11].

### 3.2. Presentation

Usually, patients present non-specific symptoms, such as irregular vaginal or postcoital bleeding.

### 3.3. Incidence

Several reports have shown an increase in the overall rate of cervical adenocarcinomas, especially in young patients [17,18,19,20,21,22,23,24]. Nevertheless, the incidence of mesonephric adenocarcinoma remains uncertain since it is often misdiagnosed and confused with more common adenocarcinomas or mistaken for benign florid mesonephric hyperplasia [25,26,27]. Clear cell carcinoma of the cervix represents a very rare variant of adenocarcinoma accounting for only four percent of all cervical carcinomas [3].

### 3.4. Age

Clear cell carcinoma and mesonephric adenocarcinoma are usually diagnosed in older patients, with a mean age of 52 years for mesonephric adenocarcinoma [14], and 71 years for clear cell carcinoma. However, a bimodal age distribution has been described for CCC, with the first peak occurring in women aged between 17 and 37 years (mean age is 26 years), and the second peak occurs in women aged between 44 and 88 years (mean age is 71 years) [28].

### 3.5. Cervical Cytology

Unlike the more common squamous epithelial carcinoma, cytology is considered to have limited benefit in the diagnosis of rare forms of cervical cancer; only a very small percentage of patients present an abnormal Pap smear. A minority of patients will present ASC-H or atypical glandular cytologic features, especially if liquid based cytology has been implemented [3,4,5].

### 3.6. HPV Genotyping

Although there are conflicting data from small series and isolated case reports, larger studies and the WHO 2020 Classification concluded that unusual variants of cervical adenocarcinomas are independent of HPV infection, with only rare exceptions, and p16 overexpression in some cases does not correlate with HPV status [9,10,15,29,30,31,32,33,34].

### 3.7. Pathogenesis

Mesonephric carcinomas are assumed to arise from mesonephric duct remnants of the female genital tract in the lateral part of the cervix [35]. Most tumors exhibit a widely infiltrative and confluent pattern of growth and usually tend to infiltrate the lower uterine segment [36,37]. Because of the widespread distribution within the cervix at the time of diagnosis, the initial site of origin in the lateral part of the cervix is often no longer apparent [27]. Similarly, primary clear cell carcinoma of the cervix shows predominantly endophytic growth and tends toward deep infiltration in the cervix and extending to uterine corpus [36,38,39].

The etiology and pathogenesis of CCC remain unclear. Intrauterine exposure to DES has been suggested as a risk factor, but it is rarely encountered [3,38,40]. Current research suggests that many factors, including cervical endometriosis, could contribute to the occurrence of CCC [3,41].

### 3.8. Pathology

The diagnosis of mesonephric adenocarcinoma is often challenging and its incidence is underestimated because of frequent misclassification. Typically, it exhibits a mixture of morphologic patterns. Therefore, they are often confused with other, more common adenocarcinomas, such as serous, clear cell or endometroid adenocarcinomas [14,36,37,42]. The typical background lesion of a mesonephric carcinoma is florid mesonephric hyperplasia, characterized by a densely eosinophilic luminal secretion [14,43]. In contrast to mesonephric hyperplasia, a mesonephric carcinoma does not have a lobular architecture and the nuclei appear cytologically malignant.

The diagnosis of clear cell carcinoma of the uterine cervix is mainly established based on histopathological examination that depicts cuboidal or hobnail cells with clear cytoplasm and markedly atypical nuclei, arranged in tubulocystic, papillary, or solid architectural patterns [3,10,38,44].

### 3.9. Immunohistochemical Study

Given its potential mimicry of other neoplasms, an immunohistochemistry can be helpful in the differential diagnosis of mesonephric adenocarcinoma (Table 1). Positive immunostaining for CD10, CK7 and calretinin along with a negative immunostaining for CEA is suggestive for a mesonephric origin. Mesonephric adenocarcinoma is also usually positive for epithelial membrane antigen (EMA) and vimentin, whereas ER/PR are usually absent [16,45,46]. PAX8 staining is usually positive in mesonephric carcinomas [27,47]. CA125 is also usually positive in mesonephric carcinoma but negative in benign mesonephric structures [27]. The immunohistochemical study is usually noncontributory to the diagnosis of clear cell carcinoma of the uterine cervix [15,48]. Park et al. [15] conducted a study on of a total of 26 cases of unusual subtypes including clear cell carcinoma (CCC, n = 9), gastric-type adenocarcinoma (GAS, n = 11), minimal deviation adenocarcinoma (MDA, n = 3), mesonephric adenocarcinoma (MSN, n = 1), serous adenocarcinoma (SER, n = 1), and malignant mixed Mullerian tumor (n = 1). They concluded that negative staining for PR and ER may serve as a general marker of endocervical neoplasia. GAS/MDA may be differentiated from all other adenocarcinomas with either positive HIK1083 stain or negative/focal p16 stain. CCC may be distinguished from all other adenocarcinomas, except MSN, with a negative CEA stain. MSN may be identified with negative CEA, ER, and PR stains [15]. Unusual variants of endocervical adenocarcinoma (UEA) are not related to HPV infection, with only rare exceptions, and p16 overexpression in non-UEA does not correlate with HPV status. p16 staining (block-type in HPV-associated and negative or mosaic-type in HPV-independent neoplasms) is much more reliable at predicting HPV status. As such, an argument can be made for undertaking p16 staining in all cervical SCC and, if staining is not block type, an HPV-independent neoplasm should be considered and HPV testing undertaken using highly sensitive molecular techniques [15,35]. The Ki-67 proliferation index is less than 1% in mesonephric hyperplasia compared to 15–20% in mesonephric carcinoma. MSN are almost always p16 negative or focally positive (nonblock-type immunoreactivity) [16,35,49].

In our case, IHC exhibited by the tumor presented certain aspects in favor of CCC (Napsin A, HNF 1B, p53) and others in favor of mesonephric adenocarcinoma (GATA3, PAX8, inhibin), rendering the differential diagnosis difficult (Table 1). Also, the first laboratory identified GATA3 expression as diffusely positive, whereas the third laboratory identified it as negative. The different GATA3 result in the third laboratory could be considered the biggest error, but it is a case that can occur in actual clinical practice. We believe this finding is important because it suggests caution in the initial diagnosis through immunostaining.

A meta-analysis of studies of dual immunocytochemical staining of Pap smears with p16/Ki-67 demonstrated a high level of sensitivity and moderate specificity for the detection of squamous cell intraepithelial lesions of the cervix and cervical cancer. Therefore, p16/Ki-67 dual-staining could represent a reliable complementary method for detecting high-grade squamous intraepithelial lesions (HSIL). However, up to date, no meta-analysis studies have evaluated the accuracy of p16/Ki-67 dual-staining for the interpretation of cases with adenocarcinoma variants of the cervix. Based on the premises that p16 and/or Ki-67 could be positive in the absence of HPV infection, dual-staining could become a useful screening test in patients with rare, non-HPV related, variants of cervical cancer [50,51]. In our case, the patient exhibited a positive dual-staining test prior to conization. Despite the report of only one case, the particularity of positive dual-staining in our patient and the theoretical probability of positive dual-staining in the absence of HPV infection render the evaluation of p16/Ki-67 in rare variants interesting and worth documenting. Also, the possibility of performing the test from the cytology sample after conization in rare adenocarcinomas should be taken into consideration.

### 3.10. Prognosis

HPV-independent cervical adenocarcinomas are typically diagnosed at advanced stages, with a higher prevalence of lymph nodes metastases and have a worse prognosis [35]. Because this disease is so rare, many of its aspects remain unclear. One multicenter study that included 34 cases of CCC from the post-DES era reported that clear cell histology in and of itself does not appear to portend a worse prognosis [52]. Given the bimodal age distribution of CCC patients, the effectiveness of fertility-preserving treatment is a key issue that requires clarification [28]. To date, only a few case reports have implied that fertility-preserving treatment is feasible in patients with early-stage CCC [53,54]. Serum CA125 has been suggested as helpful in monitoring prognosis [44].

### 3.11. Minimally Invasive or Open Surgery Approach

The Laparoscopic Approach to Cervical Cancer (LACC) study reported that laparoscopic or robot-assisted radical hysterectomy was associated with lower rates of disease-free survival and overall survival than open abdominal radical hysterectomy among women with early-stage cervical cancer [55].

However, the prospective study lacks some relevant data, such as tumor size in 1/3 of the cases and information regarding paraventricular and vaginal involvement in 7–10% of the cases, and only 39.5% of the cases reached the 4.5-year follow-up end point.

In addition, the 2019 NCCN guidelines, version 2, suggest that laparotomy, laparoscopy, or robotic laparoscopy is an acceptable radical hysterectomy approach, and laparoscopic radical hysterectomy has been demonstrated to be associated with more favorable morbidity profiles, lower costs of care, and comparable survival relative to abdominal radical hysterectomy through decades of research [56,57,58,59].

In response to the LACC study, the SUCCOR study evaluated disease-free survival in patients with stage IB1 (FIGO 2009) with cervical cancer undergoing open vs. minimally invasive radical hysterectomy, and investigated the association between protective surgical maneuvers and the risk of relapse. Despite the baseline increased risk of relapse and death compared to open surgery, protective maneuvers to avoid tumor spread at the time of colpotomy (such as vaginal closure/vaginal cuff and avoiding the uterine manipulator) have proven efficient. Minimally invasive surgery associated with risk-reducing maneuvers presented similar rates of relapse to the open surgery approach [60].

Furthermore, conization before radical surgery has been proven effective in reducing the risk of relapse and death [61].

Regarding our case, risk reducing by performing a vaginal cuff and avoiding the use of a uterine manipulator during laparoscopic hysterectomy were performed.

## 4. Conclusions

To conclude, it is important to recognize these unique variants of cervical adenocarcinoma at an early stage, as they can associate a poor clinical outcome given the usually advanced stage at the time of diagnosis. Cytological diagnosis is difficult in differentiating hyperplasia or inflammation from malignant cells in the majority of cases and discordant immunohistochemistry results between laboratories can be frequently encountered in clinical practice [62,63,64]. These findings suggest caution in the initial diagnosis through immunostaining.

The positive p16/Ki-67 dual-staining prior to LEEP, the non-specific IHC and the difficulties in establishing a diagnosis make our case an interesting one. Mesonephric adenocarcinoma and clear cell carcinoma of the cervix are so rare that many aspects remain unclear and diagnosis is often difficult. Accurate histological recognition could thus aid in the initiation of prompt therapy, thus delaying or avoiding recurrences.

Dual staining p16/Ki-67 prior to biopsy has been documented in the diagnosis of squamous and glandular lesions of the cervix, but it has not been studied so far in the diagnosis of rare adenocarcinoma variants of the cervix [63]. In this context, it could potentially be useful given the limitations of cytology and the fact that these variants are independent of HPV infection.

## Figures and Tables

**Figure 1 healthcare-10-01410-f001:**
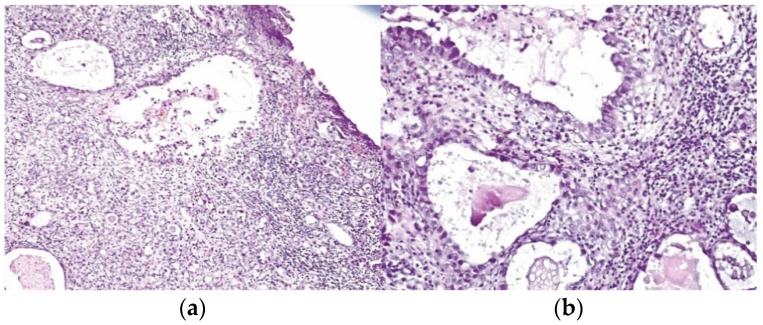
(**a**,**b**) (detail)—H&E stain—conization specimen—atypical mesonephric hyperplasia with a malignant transformation zone—mesonephric adenocarcinoma with moderate cell pleomorphism, moderate mitotic activity, without invasion of the lymphovascular space.

**Figure 2 healthcare-10-01410-f002:**
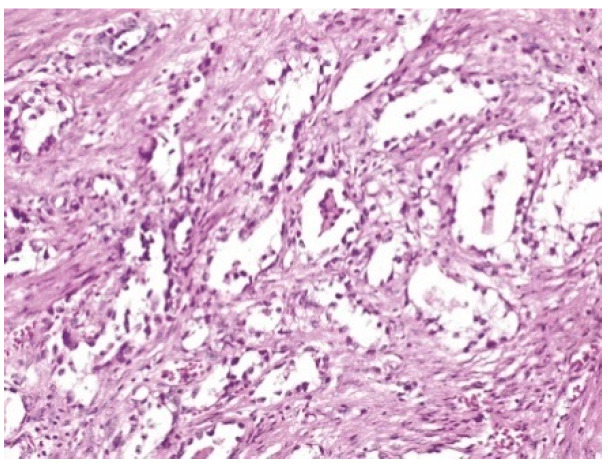
H&E stain—Vaginal wall with non-specific chronic inflammation, at the level of the subepithelial stroma we find micro focaries of mesonephric remains within the histological limits of benignity.

**Figure 3 healthcare-10-01410-f003:**
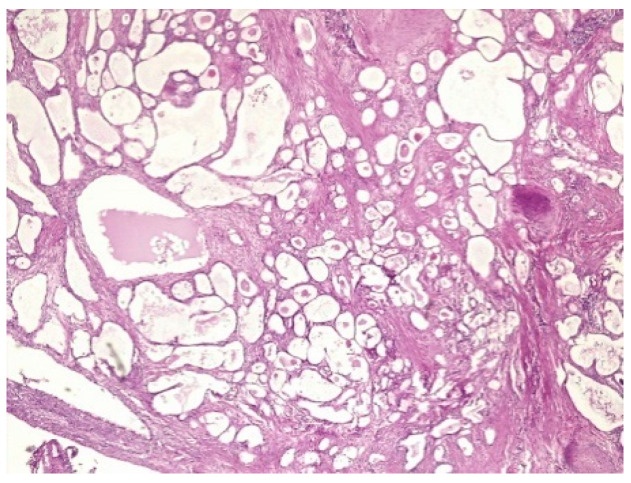
H&E stain—Trachelectomy specimen—Uterine isthmus—endocervix—upper limit of resection with benign mesonephric hyperplasia with areas of atypical mesonephric hyperplasia, showing moderate atypia.

**Figure 4 healthcare-10-01410-f004:**
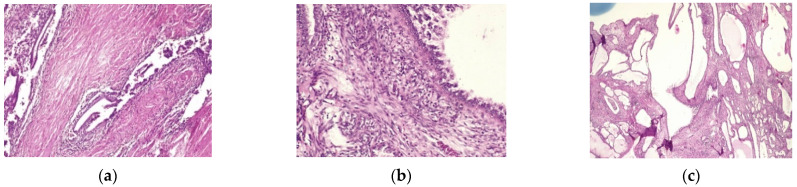
(**a**–**c**)—detail—H&E stain—Trachelectomy specimen—Cervix with previous conization—chronic ulcero-granulomatous cervicitis, condilomatous squamous epithelium; at the level of the subepithelial stroma there is a tumoral proliferation consisting of delimited tubular structures of cubic cells, non-ciliated, with moderate, eosinophilic or clear cytoplasm, presenting in the lumen an eosinophilic hyaline secretion producing a histological appearance of atypical mesonephric hyperplasia; zone of stromal invasion and malignant transformation—endocervical mesonephric adenocarcinoma with moderate cell pleomorphism and mitotic activity, intraluminal detritus, added inflammation.

**Table 1 healthcare-10-01410-t001:** Immunohistochemistry (IHC) and molecular features of the specimen compared to results presented in literature for Mesonephric hyperplasia (MH), Mesonephric adenocarcinoma and Clear-cell carcinoma of the cervix [10,11,12,13,14,15,16]; Abbreviations: pos = positive; neg = negative; ER = estrogen receptor; PR = progesterone receptor; CEA = carcinoembryonic antigen; TTF1 = thyroid transcription factor 1; First laboratory—Bioclinica, Timisoara, Romania; Second laboratory—Regina Maria, Cluj, Romania; Third laboratory—Belfast, Northern Ireland, UK.

	ER	PR	CD10	CK7	CK20	mCEA	Inhibin
First laboratory	neg	neg	neg	-	-	neg	focal pos
Second laboratory	neg	-	focal pos	pos	neg	-	-
Third laboratory	neg	neg	neg	-	-	-	-
Mesonephric adenocarcinoma	neg	neg	pos	pos	-	neg	variably pos
Mesonephric hyperplasia	neg	neg	pos	-	-	-	-
Clear-cell carcinoma	neg	neg		pos		neg	
	**TTF1**	**p53**	**GATA3**	**Ki-67**	**p16**	**PAX8**	**PAX2**
First laboratory	neg	focal pos	intense diffuse pos	12%	-	-	-
Second laboratory	-	-	-	-	-	-	-
Third laboratory	neg	-	neg	-	-	focal positive	-
Mesonephric adenocarcinoma	variably pos	neg	pos	15–20%	-	pos	pos
Mesonephric hyperplasia	-	-	pos	less than 1%	neg	pos	-
Clear-cell carcinoma	-	pos	neg	-	neg/pos	-	-
	**Calretinin**	**Napsin A**	**HNF 1B**	**ARID1A**	**Racemase**	**KRAS/** **NRAS mutation**	**AE1/AE3**
First laboratory	-	-	-	-	-	-	-
Second laboratory	-	-	-	-	-	-	focal positive
Third laboratory	-	diffuse pos	diffusepos	retention of nuclear staining	focal positive	not found	-
Mesonephric adenocarcinoma	variably pos	-	-	-	-	Canonical activating *KRAS* and *NRAS* mutations not found	-
Mesonephric hyperplasia	10%	-	-	-	-	not found	-
Clear-cell carcinoma	-	pos	pos	-	-	-	-

**Table 2 healthcare-10-01410-t002:** Schematic presentation of investigations, pathophysiological and etiological information including treatment history of the case. Abbreviations: H-SIL = high grade squamous epithelial lesion; LEEP = loop electrosurgical excision procedure; NILM = negative for intraepithelial or malignant lesions; L-SIL = Low-grade squamous intraepithelial lesion; HPV = human papilloma virus; CINtest = dual staining p16/Ki-67 MRI = magnetic resonance imaging; MDT = multidisciplinary team; HPV = human papilloma virus; CINtest = dual staining p16/Ki-67.

Five years prior to first consultation in our office	Cervical cytology: H-SIL LEEP proposed, patient refused; opted for electrocoagulation; another cervical cytology in the following 5 years: NILM
First consultation in our office	Cervical cytology: L-SIL HPV genotyping: negative CINtec test: positive Colposcopy: severe dysplastic lesion LEEP proposed
LEEP	atypical mesonephric hyperplasia with a malignant transformation zone—mesonephric adeno-carcinoma with moderate cell pleomorphism, moderate mitotic activity, without invasion of the lymphovascular space, resection limits tangential to the lesion
Abdomino-pelvic MRI after LEEP	2.2/1.6/1.6 cm formation with suspicion of malignancy which does not exceed the contour of the cervical wall, and no radiologically detectable pelvic lymphadenopathy.
Radical vaginal trachelectomy with laparoscopic pelvic lymphadenectomy	Uterine isthmus—endocervix—upper limit of resection with benign mesonephric hyperplasia with areas of atypical mesonephric hyperplasia, showing moderate atypia;
	Cervix with previous conization –appearance of atypical mesonephric hyperplasia; zone of stromal invasion and malignant transformation—endocervical mesonephric adenocarcinoma with moderate cell pleomorphism and mitotic activity, intraluminal detritus, added inflammation
	Right ilioobturator lymphadenctomy specimen—eleven lymphonodules with sinus histiocytosis, lipomatosis, no tumor metastasis; left ilioouturator lymphadenctomy specimen—seven lymphonodules with sinus histiocytosis, lipomatosis, no tumor metastasis. The stage according to FIGO classification was pT1b1 N0 Mx.
Abdomino-pelvic MRI performed after trachelectomy	Modification of the anatomy of the cervix in the postoperative context; area (10–12 mm) at the junction with the uterine body with appearance similar to the lesion described previously (MRI prior to trachelectomy)—tumor remains? No pelvic lymphadenopathy. Moderate fluid accumulations noted in the pouch of Douglas.
Final MDT decision	laparoscopic hysterectomy with vaginal cuff, left adnexectomy and transposition of right ovary

## Data Availability

Data are available from the first author, upon reasonable request.

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
