# Peer review of "New Insights in the Diagnosis of Rare Adenocarcinoma Variants of the Cervix—Case Report and Review of Literature"

_healthcare, 2022, doi:10.3390/healthcare10081410_

Round 1

Reviewer 1 Report

Clear cell carcinoma and mesonephric adenocarcinoma are known to be difficult to diagnose due to various immunohistologic staining findings. In general, GATA3, PAX8, CD10, Calretinin, HNF1B, Androgen receptor was reported to show positive expression. On the other hand, it is known that estrogen receptor (ER), progesteron receptor (PR), TTF-1, p16, Napsin A, AMACR are not expressed. However, there is still a risk of misdiagnosis by showing various morphological patterns. Therefore, it is thought that it is meaningful to introduce cases of various aspects.

To differentiate it from mesonephric hyperplasia and clear cell carcinoma, immunological staining as a representative feature of mesonephric adenocarcinoma is GATA3 +, p16 -, ER -, and frequently harbors KRAS mutations. It is thought that the biggest error is that the result of GATA3 is different from the third laboratory, but it is a case that can occur frequently in actual clinical practice. It is considered to suggest cautions in the initial diagnosis through immunostaining.

A study on dual staining p16/Ki-67 prior to biopsy has been newly presented, but it is considered that further consideration is required in the future.

Therefore, it is considered to be a paper that reflects the meaning of case reports and literature review well.

Author Response

Dear reviewer,

Thank you for taking the time to evaluate our manuscript.

You stated:

Clear cell carcinoma and mesonephric adenocarcinoma are known to be difficult to diagnose due to various immunohistologic staining findings. In general, GATA3, PAX8, CD10, Calretinin, HNF1B, Androgen receptor was reported to show positive expression. On the other hand, it is known that estrogen receptor (ER), progesteron receptor (PR), TTF-1, p16, Napsin A, AMACR are not expressed. However, there is still a risk of misdiagnosis by showing various morphological patterns. Therefore, it is thought that it is meaningful to introduce cases of various aspects.

To differentiate it from mesonephric hyperplasia and clear cell carcinoma, immunological staining as a representative feature of mesonephric adenocarcinoma is GATA3 +, p16 -, ER -, and frequently harbors KRAS mutations. It is thought that the biggest error is that the result of GATA3 is different from the third laboratory, but it is a case that can occur frequently in actual clinical practice. It is considered to suggest cautions in the initial diagnosis through immunostaining.

A study on dual staining p16/Ki-67 prior to biopsy has been newly presented, but it is considered that further consideration is required in the future.

Therefore, it is considered to be a paper that reflects the meaning of case reports and literature review well.

Reply:

We sincerely appreciate your kind remarks and the interest in our case report.

Reviewer 2 Report

Although the simple method of histological analysis by biopsy was adopted, it is a well-written case report and literature review. Two issues were raised. One suggestion is that a separate table for pathophysiological and etiological information including a treatment history should be made. The other is that it should be indicated what clinical implication has been proposed according to such a case.

Author Response

Dear reviewer,

Thank you for taking the time to evaluate our manuscript.

You stated:

Although the simple method of histological analysis by biopsy was adopted, it is a well-written case report and literature review. Two issues were raised. One suggestion is that a separate table for pathophysiological and etiological information including a treatment history should be made. The other is that it should be indicated what clinical implication has been proposed according to such a case.

Reply:

We sincerely appreciate your kind remarks and the interest in our case report. We have added the suggested table. We believe cytological diagnosis is difficult in differentiating hyperplasia or inflammation from malignant cells in the majority of cases and discordant immunohistochemistry results between laboratories can be frequently encountered in clinical practice. These findings suggest caution in the initial diagnosis through immunostaining.

Table 1 – Schematic presentation of investigations, pathophysiological and etiological information including treatment history of the case. Abbreviations: H-SIL = high grade squamous epithelial lesion; LEEP = loop electrosurgical excizion procedure; NILM = negative for intraepithelial or malignant lesions; L-SIL = Low-grade squamous intraepithelial lesion; HPV = human papilloma virus; CINtest = dual staining p16/Ki-67MRI = magnetic resonance imaging; MDT = multidisciplinary team; HPV = human papilloma virus; CINtest = dual staining p16/Ki-67

5 years prior to first consultation in our office

Cervical cytology: H-SIL

LEEP proposed, patient refused; opted for electrocoagulation; another cervical cytology in the following 5 years: NILM

First consultation in our office

Cervical cytology: L-SIL

HPV genotyping: negative

CINtec test: positive

Colposcopy: severe dysplastic lesion

LEEP proposed

LEEP

atypical mesonephric hyperplasia with a malignant transformation zone – mesonephric adeno-carcinoma with moderate cell pleomorphism, moderate mitotic activity, without invasion of the lymphovascular space, resection limits tangential to the lesion

Abdomino-pelvic MRI after LEEP

2.2/1.6/1.6 cm formation with suspicion of malignancy which does not exceed the contour of the cervical wall, and no radiologically detectable pelvic lymphadenopathy.

Radical vaginal trachelectomy with laparoscopic pelvic lymphadenectomy

Uterine isthmus – endocervix – upper limit of resection with benign mesonephric hyperplasia with areas of atypical mesonephric hyperplasia, showing moderate atypia;

Cervix with previous conization –appearance of atypical mesonephric hyperplasia; zone of stromal invasion and malignant transformation – endocervical mesonephric adenocarcinoma with moderate cell pleomorphism and mitotic activity, intraluminal detritus, added inflammation

Right ilioobturator lymphadenctomy specimen – eleven lymphonodules with sinus histiocytosis, lipomatosis, no tumor metastasis; left ilioouturator lymphadenctomy specimen – seven lymphonodules with sinus histiocytosis, lipomatosis, no tumor metastasis.

The stage according to FIGO classification was pT1b1 N0 Mx.

Abdomino-pelvic MRI performed after trachelectomy

Modification of the anatomy of the cervix in the postoperative context; area (10-12mm) at the junction with the uterine body with appearance similar to the lesion described previously (MRI prior to trachelectomy) – tumor remains? No pelvic lymphadenopathy. Moderate fluid accumulations noted in the pouch of Douglas.

Final MDT decision

laparoscopic hysterectomy with vaginal cuff, left adnexectomy and transposition of right ovary

Reviewer 3 Report

The authors describe a case of cervical adenocarcinoma unrelated to HPV. The case is described as diagnostically challenging based on the results of three different laboratory studies due to three different interpretations: mesonephric carcinoma, mesonephric duct hyperplasia, and clear cell carcinoma. Although this is an intriguing case, the value of the case report would be questioned for the following reasons:

#1. The immunohistochemical staining results are essential for diagnosis, but the fact that they vary between laboratories is problematic. For instance, the first laboratory identified GATA3 expression as diffusely positive, whereas the third laboratory identified it as negative. If the third laboratory's immunostaining results were accurate, it would be reasonable to interpret these results as clear cell carcinoma. In other words, this case does not involve a rare tumor that is intermediate between mesonephric and clear cell carcinoma and exhibits ambiguous characteristics, and we must consider the possibility that clear cell carcinoma of the uterine cervix has been misdinterpreted due to variations in the immunostaining quality of the laboratories. Based on the immunostaining performed by the third laboratory, this reviewer believes this tumor should be diagnosed as clear cell carcinoma. And if this case is a instance of clear cell carcinoma of the cervix, I do not believe it warrants a case report.

#2. I believe that the case report paper should be more concise, and that the Discussion section should not consist of a list of textbook knowledge.

Author Response

Dear reviewer,

Thank you for taking the time to evaluate our manuscript.

We sincerely appreciate your kind remarks and the interest in our case report. We hope the new version of the manuscript is more suitable to your demands. Should any more corrections be necessary, we rest at your disposal.

As an expert in the field, we consider your comments very valuable and we have modified our manuscript according to them, as follows:

You stated:

The authors describe a case of cervical adenocarcinoma unrelated to HPV. The case is described as diagnostically challenging based on the results of three different laboratory studies due to three different interpretations: mesonephric carcinoma, mesonephric duct hyperplasia, and clear cell carcinoma. Although this is an intriguing case, the value of the case report would be questioned for the following reasons:

#1. The immunohistochemical staining results are essential for diagnosis, but the fact that they vary between laboratories is problematic. For instance, the first laboratory identified GATA3 expression as diffusely positive, whereas the third laboratory identified it as negative. If the third laboratory's immunostaining results were accurate, it would be reasonable to interpret these results as clear cell carcinoma. In other words, this case does not involve a rare tumor that is intermediate between mesonephric and clear cell carcinoma and exhibits ambiguous characteristics, and we must consider the possibility that clear cell carcinoma of the uterine cervix has been misdinterpreted due to variations in the immunostaining quality of the laboratories. Based on the immunostaining performed by the third laboratory, this reviewer believes this tumor should be diagnosed as clear cell carcinoma. And if this case is a instance of clear cell carcinoma of the cervix, I do not believe it warrants a case report.

Reply:

To differentiate mesonephric adenocarcinoma from mesonephric hyperplasia and clear cell carcinoma, immunological staining has a representative feature of mesonephric adenocarcinoma: GATA3 +, p16 -, ER -, and frequently harbors KRAS mutations. We appreciate your remark that the GATA3 result is different in the third laboratory and this could be considered the biggest error, but it is a case that can occur in actual clinical practice. We believe this finding is important because it suggests caution in the initial diagnosis through immunostaining. Also, in our case, IHC exhibited by the tumor presented certain aspects in favor of CCC (Napsin A, HNF 1B, p53) and others in favor of mesonephric adenocarcinoma (GATA3, PAX8, inhibin), rendering the differential diagnosis difficult. Therefore, we consider it is meaningful to introduce cases of various aspects.

#2. I believe that the case report paper should be more concise, and that the Discussion section should not consist of a list of textbook knowledge.

Reply: We appreciate your remark. We have modified the Discussion section as suggested.

Reviewer 4 Report

The manuscript entitles “New insights in the diagnosis of rare adenocarcinoma variants of the cervix – case report and review of literature” is very interesting but unfortunately its not very novel. Its already been reviewed and published in 2015, 2019 and others.

1.    In 2019 “Application of p16/Ki-67 dual-staining cytology in cervical cancers” and author did not bother to cite the paper.

2.    In 2015 p16/Ki-67 Dual Stain Cytology for Detection of Cervical Precancer in HPV-Positive Women

3.    The figures 1, 2 , and 4(H&E staining) need to be characterized clinically with specific area and lesions.

At this point, I am not sure how author reviewed the papers and not even listed the relevant papers published in the same clinical aspects. Based on the above I am sure the this is not even categorized as case report nor even as review literature. From my view the manuscript is not complete and needs extensive amount of editing to make it publishable.

Author Response

Dear reviewer,

Thank you for taking the time to evaluate our manuscript.

We sincerely appreciate your kind remarks and the interest in our case report.

As an expert in the field, we consider your comments very valuable and we have modified our manuscript according to them, as follows:

You stated:

Extensive editing of English language and style required

Reply:

We appreciate your remark. Extensive editing of English language and style were performed by a native English speaker as suggested.

You stated:

The manuscript entitles “New insights in the diagnosis of rare adenocarcinoma variants of the cervix – case report and review of literature” is very interesting but unfortunately its not very novel. Its already been reviewed and published in 2015, 2019 and others.

  1. In 2019 “Application of p16/Ki-67 dual-staining cytology in cervical cancers” and author did not bother to cite the paper.

Reply: We appreciate your recommendation and have added the suggested references to the manuscipt. We also agree with the statement that “Cytological diagnosis of cervical glandular lesions is often difficult because of the difficulty in distinguishing inflammatory or hyperplastic changes from neoplasia

  1. Yu, L.; Fei, L.; Liu, X.; Pi, X.; Wang, L.; Chen, S.. Application of p16/Ki-67 dual-staining cytology in cervical cancers. Journal of Cancer, 2019 vol. 10,12: pp.2654-2660.
  2. Ravarino, A.; Nemolato, S.; Macciocu, E.; Fraschini, M.; Senes, G.; Faa, G.;. et al. CINtec PLUS immunocytochemistry as a tool for the cytologic diagnosis of glandular lesions of the cervix uteri. Am J Clin Pathol. 2012:138: pp.652–656
  3. Nucci MR. Symposium part III: tumor-like glandular lesions of the uterine cervix. Int J Gynecol Pathol. 2002;vol 21: pp.347–59.

We cite from 2019 “Application of p16/Ki-67 dual-staining cytology in cervical cancers”: In 40 cases of cervical adenocarcinoma, 92.5% of p16/Ki-67 dual-staining was positive, and only 1 of 16 cervical tissues without glandular lesions was dual-staining positive, suggesting that p16/Ki-67 dual-staining is a potential tool for the diagnosis of cervical glandular lesions 55.

Reference 55 Ravarino A, Nemolato S, Macciocu E, Fraschini M, Senes G, Faa G. et al. CINtec PLUS immunocytochemistry as a tool for the cytologic diagnosis of glandular lesions of the cervix uteri. Am J Clin Pathol. 2012;138:652–6  represents an interesting manuscript, but the information presented relies on glandular lesions without further specifications, as presented in the Materials and methods section, Results and Discussions: “subsequent histologic examination showed an AIS of the endocervix or AIS with early invasion.” “Our results demonstrate that CINtec PLUS immunocytochemistry may be useful also for the cytologic diagnosis of glandular lesions of the cervix uteri”. The information presented is very interesting, but our manuscript focuses on rare non-HPV related variants of cervical adenocarcinomas, as presented in the WHO and IECC classification – listed below.

You stated: 2.    In 2015 p16/Ki-67 Dual Stain Cytology for Detection of Cervical Precancer in HPV-Positive Women

Reply: We appreciate the recommendation, but unfortunately our manuscript is focused on HPV negative adenocarcinoma variants.

You stated: 3.    The figures 1, 2 , and 4(H&E staining) need to be characterized clinically with specific area and lesions.

Reply: We agree and have added the requested information in the text.

You stated: At this point, I am not sure how author reviewed the papers and not even listed the relevant papers published in the same clinical aspects. Based on the above I am sure the this is not even categorized as case report nor even as review literature. From my view the manuscript is not complete and needs extensive amount of editing to make it publishable.

Reply: We kindly appreciate your recommendations and have modified our manuscript according to them. We hope the new version of the manuscript is more suitable to your demands. Should any more corrections be necessary, we rest at your disposal.

Round 2

Reviewer 3 Report

This paper does not appear to describe a novel HPV-unassociated adenocarcinoma, but rather a practical issue that arose from a problem with interlaboratory IHC quality control. I do not believe there is sufficient diagnostic evidence to warrant reporting this case of a unique HPV-unrelated adenocarcinoma.

Author Response

Dear reviewer, we regret your lack of interest in our case, still we believe it is worth reporting and interesing given the novelty and the difficulties encountered. 

Reviewer 4 Report

No comments

Author Response

Dear reviewer,

we appreciate your previous comments that helped increase the value of our manuscript. We are pleased you have no further comments and that the current form of our manuscript is suitable to gour demands.